# Fast and Massive Pixel-Level Morphology Detection by Imaging Processing for Inkjet Printing

**DOI:** 10.3390/mi15050606

**Published:** 2024-04-30

**Authors:** Haoyang Zhang, Da Xu, Shanrong Ke, Meicong Huang, Yaling Chai, Yi Lin, Ziquan Guo, Zhong Chen

**Affiliations:** 1Department of Electronic Science, School of Electronic Science and Engineering (National Model Microelectronics College), Xiamen University, Xiang’An Campus, Xiamen 361102, China; haoyang@stu.xmu.edu.cn (H.Z.); ksr18359889778@163.com (S.K.); linyi01@stu.xmu.edu.cn (Y.L.);; 2College of Materials Science and Engineering, Central South University of Forestry and Technology, Changsha 410004, China; chaiyaling2022@163.com

**Keywords:** printed electronic devices, inkjet printing, image processing, edge detection, pixel level

## Abstract

With the rapid development of the emerging intelligent, flexible, transparent, and wearable electronic devices, such as quantum-dot-based micro light-emitting diodes (micro-LEDs), thin-film transistors (TFTs), and flexible sensors, numerous pixel-level printing technologies have emerged. Among them, inkjet printing has proven to be a useful and effective tool for consistently printing micron-level ink droplets, for instance, smaller than 50 µm, onto wearable electronic devices. However, quickly and accurately determining the printing quality, which is significant for the electronic device performance, is challenging due to the large quantity and micron size of ink droplets. Therefore, leveraging existing image processing algorithms, we have developed an effective method and software for quickly detecting the morphology of printed inks served in inkjet printing. This method is based on the edge detection technology. We believe this method can greatly meet the increasing demands for quick evaluation of print quality in inkjet printing.

## 1. Introduction

With the fast development of fifth-generation (5G) mobile communication technology, the demand for flexible, transparent, and wearable electronic devices has increased. These devices include micron-size light-emitting diode (micro-LED) displays, flexible sensors, and other flexible electronic devices [1,2]. Micro-LEDs, with their dimensions less than 50 µm, possess numerous well-known advantages such as long lifespan, fast response, low power consumption, and high brightness, making them more suitable for a variety of electronic products. These products range from smartphones and smartwatches to televisions (TVs) and extend to applications in the visible light communication (VLC), augmented reality (AR), and virtual reality (VR) among others [3,4].

The fabrication of micro-LED displays primarily follows two approaches. The first one involves combining miniaturized InGaN-based blue (B) and green (G) LEDs with AlGaInP-based red (R) LEDs to create RGB micro-LED displays [5]. The main challenge here is the mass transfer of tens of thousands of miniaturized LED chips onto target substrates, a process fraught with complexity. The second approach utilizes the color converters such as quantum dots (QDs), which are synthesized and then uniformly coated on miniaturized GaN-based blue or violet micro-LED arrays [6]. This method introduces its own set of challenges, particularly in the uniform application and patterning of QDs on the LEDs.

Recent advancements in various printing techniques, including photolithography, inkjet printing (IJP), and transfer printing (TP), have significantly contributed to the development of high-resolution electronic devices, such as QD-based micro-LEDs [6,7]. In particular, IJP technology has gained prominence for its role in producing low-cost, large-scale, lightweight, optically transparent, and scalable electronic devices, encompassing a broad range of applications from thin-film transistors (TFTs) to supercapacitors [2,6].

Despite its advantages, IJP still faces several issues like the coffee-ring effect, non-uniform films, challenges in large-scale fabrication, and surface roughness [8]. The importance of effective tools and methods for rapidly assessing the physical size, morphology of deposited droplets, and the overall printing quality of IJP is becoming increasingly crucial, especially given its widespread future applications. Uniformity and high resolution in the printed patterns are essential for enhancing the device stability. Recent studies, such as those by S. Shi et al. using microscale fluorescence spectroscopy (MFS) for the uniformity assessment of quantum-dot pixels [9] and by Behrman et al. through photoluminescence (PL) and cathodoluminescence (CL) imaging for the defect identification in micro-LEDs [10], highlight the ongoing efforts in this area. Additionally, H. Zhang et al.’s proposal of a machine learning approach for optimizing aerosol jet printing (AJP) based on droplet morphology underscores the innovative directions in addressing these challenges [11]. Yet, this literature reveals a scarcity of fast and extensive techniques for the ink droplet detection in the IJP, considering the imaging processing algorithms.

In response to this gap, our study aims to introduce a rapid and large-scale methodology for detecting the morphology of ink droplets during the IJP process by using the image processing algorithms. This approach is poised to significantly reduce the time required for mass detection and quality assessment of inkjet printing, which is pivotal for practical applications. To achieve high-resolution images of printed ink droplets, a metallographic microscope equipped with a charge-coupled device (CCD) camera is employed, commonly used in the optical inspection of LEDs [12].

## 2. Experimental

### 2.1. Intrinsic Factors

Several important intrinsic factors would affect the morphology of ink droplets during the inkjet printing process. For instance, the viscosity, density, and surface tension of the ink all contribute to a *Z* value, which is the inverse of the dimensionless Ohnesorge Number (*O_h_*), like [9]
(1)$$Z=\frac{1}{O_{h}}$$

For stable droplet jetting, it is advisable, based on experience, to maintain the *Z* value within the range of (1, 14). Additionally, external factors such as the electrical waveform parameters, including bias voltage, pulse width, and voltage amplitude, can influence the physical size and morphology of ink droplets. The diameter and height of the nozzle to the substrates also play significant roles in the jetting of ink droplets. Effective control over these mentioned conditions can lead to the generation of relatively regular and uniform ink droplets, minimizing the occurrence of satellite droplets during the printing process.

### 2.2. Experimental Conditions

Figure 1A shows the schematic diagram of this inkjet printing as electrohydrodynamically (EHD) printing. The inkjet printing experiment is carried out by using the ultra-high resolution material deposition printer, denoted as SIJ-350, which is manufactured by SIJ Technology (Tsukuba, Japan). The printer is equipped with a nozzle diameter of 5 µm, allowing for the precise deposition of ink droplets with feature sizes of less than 20 µm, which is notably finer than the capabilities of standard inkjet printer nozzles. The first inks used in this study consist of red perovskite quantum dots (QDs) and a UV-curable polymer, suitable for the fabrication of full-color QD-based micro-LEDs; the second ink used in this study is Ag inks. The substrate for printing is a standard transparent glass substrate, with the dimension of 5 cm in both length and width and a thickness of 1 mm. The high-resolution images of the printed ink droplets are captured by using a metallographic microscope (RX50M, Ningbo Sunny Instruments Co., Ltd., Ningbo, China) with magnification ranging from 10× to 1000×, coupled with a CCD camera.

During the configuration of the SIJ-350 software v1.0, the electrical waveform used to drive the jetting is set to a 75% square wave, with a frequency of 1000 Hz. The split amplitude is set at 150 V and the split bias at 100 V, with a split speed of 0.2 mm/s. Figure 1A also displays the printed red perovskite QD ink droplets alongside Ag ink droplets, arranged in a 4 × 4 array. It is evident that the coffee-ring phenomenon is observable in the QD ink droplets, whereas this effect is absent in the Ag ink droplets, which maintain a regular and uniform circular shape.

### 2.3. Image Edge Detection Algorithms

The image detection algorithms often involve edge detection techniques, where pixel values in an image undergo abrupt changes. These techniques are widely employed in image segmentation tasks. The realm of image processing features numerous edge detection (or gradient) operators, including first-order edge operators such as the Sobel, Scharr, Roberts, Prewitt, and Kirsch operators, as well as second-order edge operators like the Laplacian and Canny operators.

The expression for the Laplacian operator is as follows:(2)$$\nabla^{2}f(x,y)=f(x+1,y)+f(x-1,y)+f(x,y-1)+f(x,y+1)-4f(x,y)$$

The expression for the Laplacian enhancement operator is as follows:(3)$$\begin{array}{l} g(x,y)=f(x,y)-\nabla^{2}f(x,y)\\ =5f(x,y)-[f(x+1,y)+f(x-1,y)+f(x,y+1)+f(x,y-1)] \end{array}$$

Figure 1B presents a flow chart illustrating the edge detection process utilized in this study.

1. Initially, an image captured by a high-precision camera is input;

2. The image then undergoes grayscale processing to simplify analysis;

3. Subsequently, the Gaussian fitting is applied to mitigate noise in the image;

4. Importantly, threshold processing is essential to enhance the clarity of the edges in the image. Without this step, there would be a lot of noise, characterized by rough, indistinct, and discontinuous edges, leading to inaccurate positioning;

5. Finally, edge processing is performed on the images using an edge operator to highlight and delineate the edges within the images.

The edge detection algorithms employed in this study are particularly tailored for high-contrast and noise-minimal imaging conditions which are typical of inkjet printing assessments. The adaptive thresholding technique is utilized to dynamically adjust edge detection sensitivity, mitigating false edge recognition in varied lighting and background conditions. While this methodology is optimized for the SIJ-350 printer system, the foundational principles are applicable across different systems after minimal adjustments, ensuring broad applicability in diverse imaging environments.

## 3. Results and Discussion

In this section, the outcomes of the proposed methods are analyzed. Figure 2A presents the original Ag printed pattern, which is identified as a bimodal image due to the two prominent peaks in its grayscale histogram, as clearly depicted in Figure 2B. This characteristic makes the Otsu threshold method [13] as a binary image segmentation algorithm, particularly apt for processing such types of images. The edge-delineated image, as seen in Figure 2A, exhibits notable continuity and clarity.

In order to objectively evaluate the detection performance, three image quality assessment indicators, Peak Signal to Noise Ratio (PSNR), Structural Similarity Index (SSIM), and Mean Squared Error (MSE), are used to verify the detection accuracy of the algorithm, as follows:(4)$$MSE=\frac{1}{M\times N}\sum_{i=1}^{M}\sum_{j=1}^{N}{\left[g(i,j)-f(i,j)\right]}^{2}$$
where *M* and *N* are the number of pixels in the length and width of the image, respectively, and *g*(*i, j*) and *f*(*i, j*) are the grayscale values at that point before and after processing. And
(5)$$PSNR=20\log_{10}(\frac{MAX_{I}}{\sqrt{MSE}})$$
where *MAX_I_* represents the maximum pixel value of the image, which is 255, and *MSE* represents the mean squared error between the original image and the detection image.
(6)$$SSIM(x,y)=\frac{\left(2\mu_{x}\mu_{y}+C_{1})(2\sigma_{xy}+C_{2}\right)}{\left(\mu_{x}^{2}+\mu_{y}^{2}+C_{1})(\sigma_{x}^{2}+\sigma_{y}^{2}+C_{2}\right)}$$
where *x* and *y* represent the pixel values of two images, *μ_x_* and *μ_y_* represent their average luminance, *σ_x_*^2^ and *σ_y_*^2^ are the variances of *x* and *y,* respectively, *σ_xy_* is the covariance between *x* and *y*, and *C*_1_ and *C*_2_ are small constants introduced to avoid division by zero, typically depending on the dynamic range of the image data.

To ensure a fair and unbiased evaluation, we randomly selected 10 images from our dataset and analyzed them by using our method, calculating the PSNR, MSE, and SSIM metrics accordingly. For comparison, images processed solely with the Laplacian operator for edge detection show an average PSNR of 28.2436, a higher MSE of 97.4352, and a significantly lower SSIM of 0.0170, indicating lesser image reconstruction quality and higher error levels. In contrast, our method achieves a PSNR of 30.4700, MSE at 58.3545, and SSIM at 0.1339. These results not only demonstrate an improvement over the basic Laplacian approach in terms of noise resilience and error reduction but also highlight the enhanced structural fidelity that our method offers, as reflected in higher SSIM values.

This analysis underscores the good consistency and stability of our evaluation metrics across the randomly selected image samples, providing a comprehensive view of the method’s performance. The comparison confirms the accuracy and output consistency of our method, significantly outperforming the Laplacian operator in crucial aspects. Our findings support the efficacy of the method and indicate substantial potential for improvement, particularly in areas of structural fidelity as evidenced by the SSIM metric.

In this study, we evaluate the Laplacian operator (Figure 2C) performance against other operators in terms of processing time, with the findings presented in Figure 3. The Laplacian operator demonstrates the most efficient processing time at 3.95 s, marked with a five-pointed star. Consequently, in this work, we employ a combination of the Otsu threshold method and the Laplacian operator for processing images obtained from the inkjet printing.

Here, we first use the Hough circle detection algorithm to identify the center of circular edge of ink droplets and obtain the position of edge pixels in the circle. Then, we calculate the distance between each edge pixel and the center of circle separately and take the average value to obtain average diameter of each circle. However, since the diameter is measured in pixels, we need to select the suitable scaling factor (*SF*) to convert the pixel unit in the image to the actual unit (µm) as
(7)$$SF=\frac{L_{pixel}}{L_{real}}$$

The scaling factor is adjusted according to the magnification level of the microscope, being set at 0.3 for 10×, 1.2 for 50×, and 3 for 100× magnification levels, respectively, by experience. Figure 4A presents the edge detection outcomes obtained under the 10× microscope magnification, where the edges resulting from the analysis are superimposed on the original image to provide a clear visualization of the detected boundaries. After removing some bad points, a total of 437 pixels points are processed in the figure. Due to the large number of pixel points and low resolution, the detection results are not precise enough. Figure 4B also shows the statistical histogram of the detected average diameter (*d*) of 437 pixels points. We can see that most pixel points have a diameter of about *d* = 14–16 µm. Figure 5A shows the edge detection results obtained at a magnification of 20× of the microscope, where the analyzed edges are superimposed on the original image to provide a clear visualization of the detected boundaries. A total of 130 pixels are processed in the figure. Due to the moderate number of pixels, more accurate detection results can be obtained by processing fewer pixels at once. Figure 5B also shows the statistical histogram of the average diameter (*d*) of 130 detected pixels. We can see that the diameter of most pixels is approximately *d* = 12–13 µm, proving that this proposed method is useful for quick detection of droplet morphology and physical size in inkjet printing.

## 4. Software Design

In this project, we have chosen the Qt Creator as the development tool and C++ as the programming language for implementing backend algorithms. For interface development, the Qt Designer will be used to achieve the desired GUI design. To facilitate effective integration between the interface (frontend) and functionality (backend), we will encapsulate the functional routines written in C++ into dynamic-link libraries (DLLs), allowing them to be called from C#.

The main interface of this software is shown in Figure 6A. In it, the top left corner of the software contains the system’s menu bar and the central part is the original image display area after the software has successfully imported the original images.

The system menu bar includes five options: File, Set, Method, Report, and Help. The File option allows users to import the original inkjet-printed image data that need to be analyzed. The Set option helps users configure the scale, entering the conversion ratio between pixel size and actual size into the system, thus achieving the conversion between image pixels and actual dimensions to meet real production needs. The Method option offers a variety of edge detection methods suitable for inkjet-printed images under different conditions. Through the Report option, this software provides users with image analysis results and statistical information, facilitating further use or processing by the users. The Help section assists users in understanding how to operate the software and provides a brief introduction to the system.

After drawing lines on the original image according to the scale, clicking the Set button on the menu bar will prompt a scale setting dialog box, as shown in Figure 6B. Upon entering the actual length and unit of the scale based on real data, the system backend will automatically calculate the ratio between image pixels and actual dimensions. This ratio is then used in subsequent analyses to convert sizes, facilitating the generation and export of inspection reports.

Once the scale and detection method is set, clicking the Report button on the menu bar will open the report window, as illustrated in Figure 6C. Within the inspection report dialog, two report formats are available for the user to choose from, a txt file and a jpg file, as depicted in Figure 6C, respectively. Users can export the inspection report file information according to their needs.

## 5. Conclusions

In summary, based on the image processing algorithms related to edge detection operators, via the metallographic microscope followed by a CCD camera and SIJ-350 devices, we carry out a study on the massive detection of morphology of printed ink droplets in the inkjet printing. The combination of the Hough algorithm, Otsu, and Laplacian operator is adopted to process the images of printing. Finally, a scaling factor (SF) is used to convert the pixel unit in the image to the actual unit of printed ink droplets. Finally, software related to the proposed method has been well designed for conveniently processing the images from inkjet printing. We believe that this proposed method and related self-designing software are useful for the quick detection of morphology and size of printed ink droplets in inkjet printing. Future works will focus on the coffee ring detection of printed quantum dots and printed Ag lines.

## Figures and Tables

**Figure 1 micromachines-15-00606-f001:**
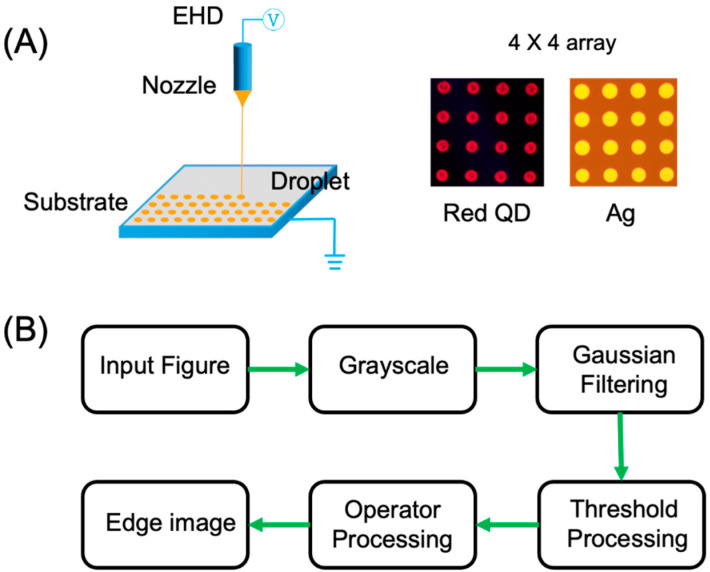
(**A**) The schematic diagram of inkjet printing. The image of printed QD ink droplets and Ag ink droplets, arranged in a 4 × 4 array. (**B**) The flow chart of edge detection in this work.

**Figure 2 micromachines-15-00606-f002:**
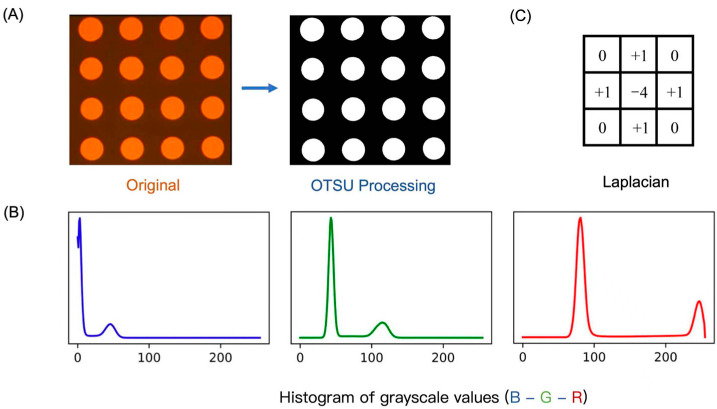
(**A**) The original Ag printed pattern and after the Otsu image processing. (**B**) The histogram of grayscale values of R, G, and B. Two prominent peaks in its grayscale histogram can be found. (**C**) The Laplacian operator.

**Figure 3 micromachines-15-00606-f003:**
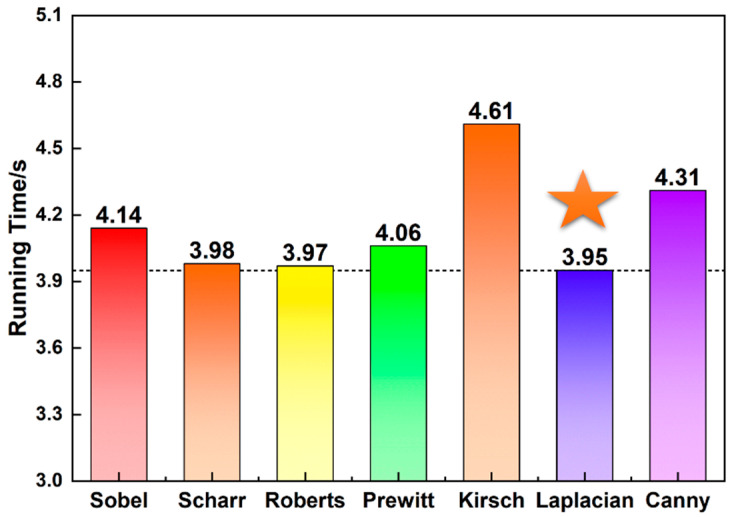
The running time of seven typical edge operators. The Laplacian operator demonstrates the most efficient processing time at 3.95 s, marked with a five-pointed star.

**Figure 4 micromachines-15-00606-f004:**
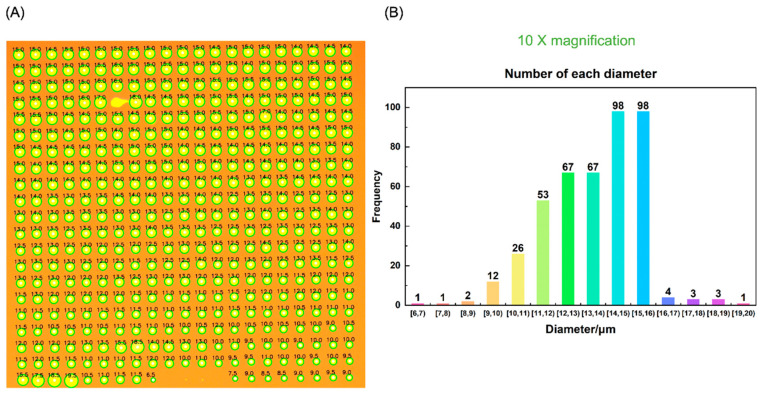
(**A**) The Hough + Laplacian + Otsu detection results and (**B**) the statistical histogram of 10× magnification.

**Figure 5 micromachines-15-00606-f005:**
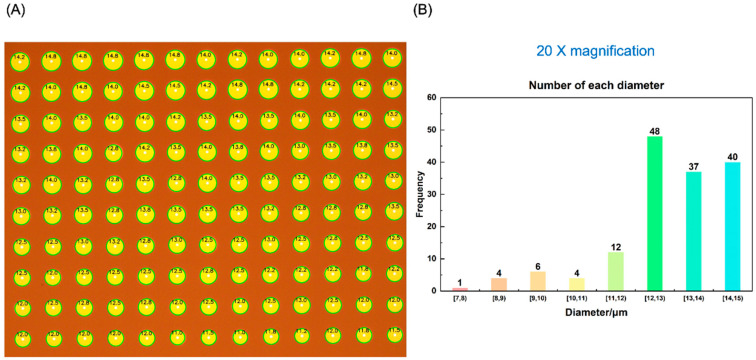
(**A**) The Hough + Laplacian + Otsu detection results and (**B**) the statistical histogram of 20× magnification.

**Figure 6 micromachines-15-00606-f006:**
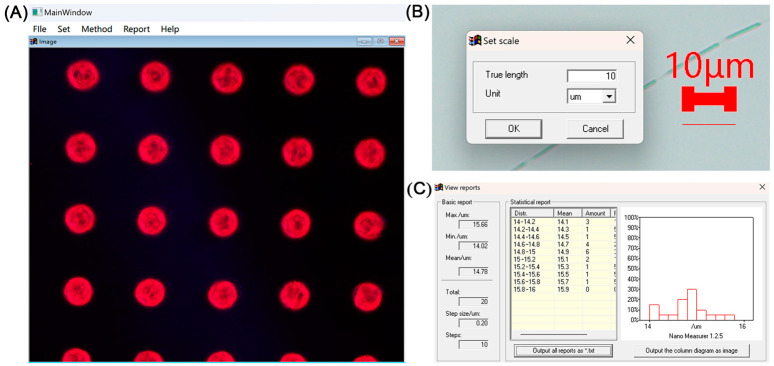
(**A**) Main interface of the system. (**B**) Interface for setting the scale. (**C**) Interface of the detection report after image processing.

## Data Availability

Dataset available on request from the authors.

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
