# Peer review of "Fast and Massive Pixel-Level Morphology Detection by Imaging Processing for Inkjet Printing"

_micromachines, 2024, doi:10.3390/mi15050606_

Round 1

Reviewer 1 Report

Comments and Suggestions for Authors

The authors designed a method and software that efficiently identifies the morphology of inks utilized in inkjet printing by applying various image processing algorithms that are already in existence. Some of my comments are given below:

  1. What is the detection accuracy of the proposed method and its RSD value?
  2. The authors should describe the repeatability performance of their method.
  3. The author should include some comparison of their proposed method with others.

Reviewer 2 Report

Comments and Suggestions for Authors

The manuscript entitled “Fast and massive pixel-level morphology detection by imaging processing for inkjet printing” contains an interesting and systematic research paper on the development of an effective method based on edge detection technology for fast detection of the morphology of printed inks used in inkjet printing. The topic is original and relevant to the field as inkjet printing, among other technologies, has contributed significantly to the development of high-resolution electronic devices and has become known for its role in the production of low-cost, largescale, lightweight, optically transparent and scalable electronic devices.

The manuscript is well organized, the authors have applied the scientific methods and they are adequately described. The results are clearly described and the discussion section is well presented. At the end of the manuscript, the authors state that the results presented in this manuscript are useful for rapid detection of the morphology and size of printed ink droplets in inkjet printing.

The references cited in the manuscript are recent, mostly from the last five years. Most figures and images are appropriate and clearly presented. Data presented in charts are appropriately presented and easy to interpret and understand. Most of the concluding observations are written down in the discussions that I consider appropriate for this research.

Although the paper sounds very interesting, it is lacking on certain points. It is said that a high resolution SIJ-350 printer was used. Is it an Epson printer or another? No manufacturer is specified. The inks and substrate used are not specified. It is known that the interaction between the ink and the substrate plays the most important role in forming the shape of the droplet on the substrate.

It also needs to be clarified whether there are certain limitations in the use of edge detection algorithms. How is the threshold at the edge of the droplet managed and can it be used only in this proposed system or also in other systems?

Also, the list of references is quite short. Is this not such an advanced field that there are not more authors who have worked on it? There is a need to expand the list with references that is in this area.

I suggest a major revision taking into account the shortcomings mentioned.

Round 2

Reviewer 1 Report

Comments and Suggestions for Authors

The authors improved their manuscript correctly; I believe the manuscript can be accepted for publishing in the Micromachines journal in the present form.

Reviewer 2 Report

Comments and Suggestions for Authors

The reference list has not been improved in the revised paper.

Without these corrections, the paper cannot be accepted.